

# Storylines of Future Drought in the Face of Uncertain Rainfall Projections: a New Zealand Case Study

Hamish Lewis[1,2], Luke J. Harrington[1], Peter B. Gibson[2], and Neelesh Rampal[2]

[1]School of Science, University of Waikato, Hamilton, New Zealand
[2]National Institute of Water and Atmospheric Research (NIWA), New Zealand

**Correspondence:** Hamish Lewis (hamish.lewis@waikato.ac.nz)

**Abstract.** Robust increases in temperatures will occur globally across the twenty-first century; however, for some regions, sign changes in rainfall remain uncertain. Navigating this uncertainty is crucial in addressing drought-related challenges faced by climate-exposed sectors. New Zealand represents these challenges, exhibiting significant model uncertainty in warm-season rainfall change. Here, we examine how temperature-driven increases in potential evapotranspiration interact with contrasting

storylines of future rainfall to explore drought outcomes for New Zealand. In our drying storyline, we find that increasing temperatures accompanied by less rainfall bring forward the onset of drought several months and delay its termination. In the wetting storyline, increases in rainfall partly offset the temperature-induced drying effect, leading to minor reductions in soil moisture. Examining extreme years, the average hydrological year in the future becomes comparable to the driest years of the current climate, while the worst future events exhibit unprecedented drought severity.

## 1 Introduction

The environmental and economic risk posed by severe drought events is projected to rise in the twenty-first century as global temperatures increase (Lesk et al., 2016; Naumann et al., 2021; Singh et al., 2022). While models from the Coupled Model Intercomparison Project phase 6 (CMIP6, Eyring et al. (2016)) exhibit a large range in the magnitude of the 'likely' (66% probability) global mean surface temperature response (2.3-4.5°C) to a doubling of CO2 concentrations (Sherwood et al.,

2020), the sign of mean rainfall changes remains uncertain for many regions of the world (Lee et al., 2021). Predictions of warming-induced changes in global rainfall patterns over subtropical and mid-latitude land masses vary significantly (Byrne and O'Gorman, 2015; Allan et al., 2020), partly due to different circulation changes and horizontal gradients of change in surface air temperature. Some climate models can overestimate the future response to jet changes (Simpson et al., 2014) due to an equatorward biased climatological jet position (Vallis et al., 2015; Curtis et al., 2020), which enhances the rainfall

response to future forcings. Future changes in ENSO have the ability to impose changes in regional rainfall outside of the tropics (McGregor et al., 2022), as well as inducing changes in other large-scale features (Haarsma and Selten, 2012; Teng and Branstator, 2019; Wang et al., 2022). Model internal variability can also introduce irreducible uncertainty for future climate projections (Deser et al., 2012; Lehner et al., 2020).





The above factors are exemplified when looking at projected rainfall changes over New Zealand. Here, previous research
has found model uncertainty and substantive internal variability both contribute to highly uncertain projections of warm-
season rainfall change over the twenty-first century (Gibson et al., 2024a). Subsequently, there is considerable uncertainty
about whether future rainfall changes will alleviate or exacerbate future drought events already strengthened by background
warming.

Past efforts to understand future drought risk across New Zealand also reveal mixed results. When examining the risks of
meteorological drought only, Harrington et al. (2016) found the synoptic conditions responsible for the particularly extreme
rainfall deficits across the North Island in early 2013 were more likely to occur in the recent climate than corresponding model
simulations of the late 1800s. This was supported by Gibson et al. (2016) who found circulation types linked to low rainfall
across the country were projected to occur more often in the twenty-first century under a high-warming (RCP8.5) scenario.
However, previous assessments of changing agricultural drought risk across New Zealand (Mullan et al., 2018; Sood and
Mullan, 2020) have instead largely reflected climatological patterns of rainfall: wet regions in the south-west of the country
were expected to experience a reduction in drought risk over the twenty-first century; the drier regions of the country were
likely to experience an increase in drought; while many parts of the North Island exhibited no clear trend either way.

A key drawback of previous research examining agricultural drought with downscaled climate projections in New Zealand
has been the focus on communicating future projections based on the mean of the multi-model ensemble only (Mullan et al.,
2018; Clark et al., 2011; Mullan et al., 2005). However, given the rainfall-related uncertainty across models (Gibson et al.,
2016), physically plausible but extremely diverse outcomes often end up being averaged out when looking only at ensemble-
mean outcomes. To overcome this, we focus here on two different storyline scenarios (Shepherd et al., 2018) to investigate how
warm-season rainfall changes could affect the severity of future droughts across New Zealand. In the first storyline, we consider
a "wetting" scenario, where warming-driven increases in potential evapotranspiration are offset by increases in rainfall. The
second storyline we consider is the "drying" scenario, where rainfall decreases in a warming climate and exacerbates the soil
moisture loss from warming. Both storylines are well represented by at least one dynamically downscaled CMIP6 model in
recently completed regional climate projections for New Zealand (Gibson et al., 2024b). Past research has found the diversity of
future outcomes in this ensemble of regional climate projections likely spans the plausible range of uncertainty in temperature
and rainfall changes for a given future scenario (in our case, SSP370). Thus, we are confident that we successfully explore a
fuller representation of the model uncertainty space with these two storylines.

This study primarily focuses on defining drought using a single-layer soil water balance model, driven by daily rainfall and
Potential Evapotranspiration (PET) derived from the output of dynamically downscaled regional climate model projections.
This water balance model has been extensively used for drought projections in New Zealand in previous studies (Mullan et al.,
2005; Clark et al., 2011; Mullan et al., 2018), real-time drought monitoring (Mol et al., 2017), and for soil moisture estimates in
New Zealand most comprehensive gridded station data product (Tait et al., 2006, 2012). We thus adopt this approach primarily
to ensure consistency with past approaches but also because this simple model of soil moisture can provide us with a clear
understanding of the first-order effects of temperature and rainfall change on future drought.





The remainder of this paper is structured as follows. In Section 2, we first discuss the regional climate model (RCM) data used in this analysis and the water balance model used to calculate drought metrics. In Section 3, we examine future drought across

our two storylines and discuss them in the context of the other CMIP6 models used to produce regional climate projections for New Zealand. We discuss the limitations of this work in Section 4 and summarise our main findings in Section 5.

## 2    Data and Methods

### 2.1    Regional Climate Model Data

We examine historical and future drought in six models from the CMIP6 ensemble, which have been dynamically downscaled

from their native resolutions to a resolution of 12 km over New Zealand (Gibson et al., 2024b). These six models were chosen because of their performance over the historical period using processes-based metrics (i.e. jet position), model independence, and spread in equilibrium climate sensitivity (Gibson et al., 2024b). Dynamical downscaling was performed using the Conformal Cubic Atmospheric Model (CCAM, McGregor and Dix, 2008) which implements a variable resolution conformal-cubic grid to provide an enhanced resolution over the area of interest (New Zealand and its surrounding ocean), accompanied by a

relatively high resolution (12-35km) over the wider South Pacific. Three GCMs (ACCESS-CM2, EC-Earth3, NorESM2-MM) were downscaled with CCAM through spectral nudging to the host GCM's atmospheric fields, sea surface temperatures (SSTs), and sea ice concentrations (SICs). Three other GCMs (AWI-CM-1-1-MR, CNRM-CM6-1, GFDL-ESM4) were downscaled using an "AMIP" configuration, driving CCAM with bias-corrected host model SSTs and SICs. We consider two time periods in this analysis, a historical period (1985-2014) and a future period (2070-2099) from a relatively high emissions scenario

SSP370.

### 2.2    Water Balance Model and Drought Metrics

We use daily mean values from the RCM as input to a simple water balance model to estimate the soil moisture state, actual evapotranspiration (AET) and the potential evapotranspiration deficit (PED). The soil moisture content (SM) is given by the following soil water budget equation (A. S. Porteous and Salinger, 1994):

$$\mathrm{SM}(Day) = \mathrm{SM}(Day-1) + \mathrm{Precip}(Day) - \mathrm{AET}(Day) \tag{1}$$

Soil moisture deficit (SMD) is defined as the amount of water the soil is short of field capacity (FC): $\mathrm{SMD} = \mathrm{SM} - \mathrm{FC}$. The FC is determined by the soil characteristics; an FC of 150mm is typical for New Zealand's silt-loam soils (A. S. Porteous and Salinger, 1994). With enough available soil moisture AET is able to meet atmospheric demand and equal PET, which is calculated using the FAO-56 Penman-Monteith method (Allen et al., 1998; Prudhomme and Williamson, 2013) for an

idealized grass reference crop of height 0.12 m and an albedo of 0.23 (Allen et al., 1998; Vremec et al., 2023). As soil moisture decreases, there is no longer enough readily available water to meet the atmospheric demand of PET. Based on Porteous &



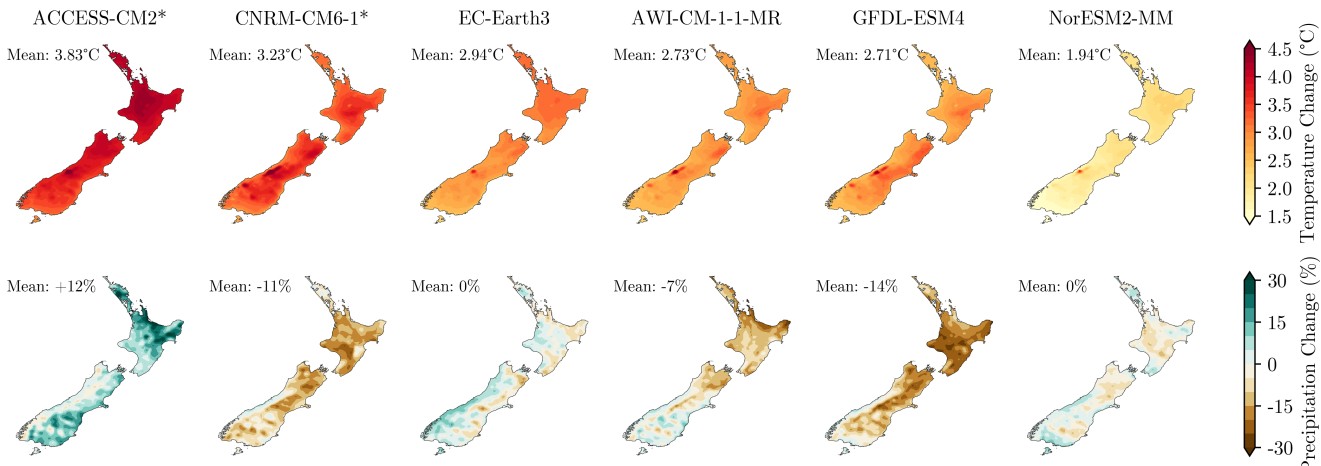

**Figure 1.** ONDJFM Temperature and rainfall changes between the historical (1985-2014) and future (2070-2099) periods examined in this study, for the six models in our ensemble ordered from warmest to coolest. The models which encapsulate the wetting (ACCESS-CM2) and drying (CNRM-CM6-1) storylines are highlighted with an asterisk. The mean change over the New Zealand land surface across low lying areas (elevation < 500m) is presented for each model.

Salingers (1994) single-layer water balance model, if half or more of the FC is available, PET and AET are set equal, and if less than half of the FC is available, AET reduces linearly proportional to PET:

$$
\text{AET} = \begin{cases} \text{PET} & \text{SMD} > -75\text{mm} \\ \text{PET}(1 + \frac{\text{FC}/2 + \text{SMD}}{\text{FC}/2}) & \text{SMD} < -75\text{mm} \end{cases}
\tag{2}
$$

In addition to examining drought with SMD, we also examine PED, defined as the difference between PET and AET. We focus on accumulated changes in PED over the June-July water year. Positive values of PED imply that the soil is too dry to meet water demand, and plants do not have sufficient access to moisture for optimal growth.

## 3   Results

By design, the six dynamically downscaled climate models have considerably different warming rates. We analyse changes in
ONDJFM (October - March) as this is the period over which droughts generally intensify, peak, and begin to recover. Surface air temperatures between the historical and future periods increase between 1.94-3.83°C, on average across New Zealand. These temperature changes are shown in Figure 1, where we also present the change in ONDJFM rainfall as a percentage. Across the models, only CNRM-CM6-1 and GFDL-ESM4 share a consistent sign of change across the majority of New Zealand, with the other models having more varied spatial patterns of rainfall change. When examining future warm-season mean changes
in temperature and rainfall across this ensemble, two models stand out as good candidates to represent contrasting storylines




of future drought in New Zealand. ACCESS-CM2 has the highest warming of all the models within the ensemble (3.83°C), and has mean rainfall increasing across all regions of New Zealand. Conversely, CNRM-CM6-1 has the second highest future warming (3.23°C) while rainfall decreases in all regions. These two model storylines of "wetting" and "drying" represent either end of the distribution of possible rainfall changes which could affect drought: thus, we focus on these two models for

the remainder of the analysis (equivalent figures for all other models are shown in the supplementary material).

Differences in regional projections of precipitation in climate models may arise for many different reasons (Gibson et al., 2024a). Figures 2 and 3 examine changes in MSLP, 500 hPa geopotential heights, and 850 hPa specific humidity, alongside precipitation, in both the early (OND) and late (JFM) dry season to investigate the drivers of precipitation projections in our two storyline models (equivalent figures and explanations for the other models in the ensemble are presented in the supplementary

material). Shown in Figure 2 the projection of drying in CNRM-CM6-1 is mostly driven by the drying in OND. This early season drying over New Zealand appears to be driven by the increase in high pressure directly over New Zealand (comparable changes are seen in other models with drying signals, GFDL-ESM4 and AWI-CM-1-1-MR). Synoptically ACCESS-CM2 (shown in Figure 3) is very similar to other models in the ensemble which show little to no increases in precipitation (EC-Earth3, NorESM2-MM), with a region of increased high pressure occurring further to the east of New Zealand in OND than

the drying models. However, increased moisture availability particularly in the late dry season (i.e. JFM) coincidences with increased precipitation over this period. The future position of the jet stream is similar across all ensemble members, as well as all members showing a tendency towards an El Niño state in the future mean that these features are unlikely to contribute to the precipitation change differences across the ensemble. Interpretation of the drivers for the other models in the ensemble are provided in the supplementary material.

In our two storyline models, all else being equal, the large degree of warming will increase PET and thus also increase the drying rate of soil moisture. This drying will be somewhat mitigated in ACCESS-CM2 due to future increases in rainfall and exacerbated in CNRM-CM6-1 due to decreases in rainfall. Indeed, this is confirmed when examining future changes to cumulative PET and AET across a full hydrological (July-June) year for six low-lying agriculturally important regions around New Zealand, Figure 4 (a), where changes in PET are correlated with mean temperature (b), and where changes in AET are

correlated with rainfall (c) across all 6 GCMs.

Figure 5 examines PET, AET and SMD in both historical and future periods across our six case study locations for the two selected models. In the top and third row of Figure 5, increases in PET due to the effect of rising future temperatures can be seen in both models (Allen et al., 1998). In ACCESS-CM2, there is a distinct increase in AET from January onwards in the Far North, Waikato, and Hawkes bay. This increase is likely due to two factors. First, the slightly smaller SMD shown in the

second row of Figure 5, which is due to future increases in mean rainfall. Second, the increase in PET while the SMD remains at a similar level, as is the case for the remainder of the regions in ACCESS-CM2 (Manawatu, Canterbury, and Southland). In the CNRM-CM6-1 model the reduction in Spring rainfall causes the SMD to reach its summertime values far sooner in the Far North, Waikato, Manawatu and Canterbury, causing a large reduction in AET over this period. Decreases in Summer and





**Figure 2.** Changes in large scale conditions between 1985-2014 (historical) and 2070-2099 (SSP370) for the CNRM-CM6-1. (a-b) Changes in mean sea level pressure (MSLP) in (a) (OND) and (b) (JFM) between 1985-2014 in the Historical period and 2070-2099 in SSP370. (c-d) Similarly to (a-b) but for changes in geopotentail heigh ($Z$) at 500 hpa. (e-f) Similarly to (a-b) but for precipitation. (g-h) Similarly to (a-b) but for specific humidity (Q) at 850 hPa.





**Figure 3.** Changes in large scale conditions between 1985-2014 (historical) and 2070-2099 (SSP370) for the ACCESS-CM2 model. (a-b) Changes in mean sea level pressure (MSLP) in (a) (OND) and (b) (JFM) between 1985-2014 in the Historical period and 2070-2099 in SSP370. (c-d) Similarly to (a-b) but for changes in geopotentail heigh ($Z$) at 500 hpa. (e-f) Similarly to (a-b) but for precipitation. (g-h) Similarly to (a-b) but for specific humidity (Q) at 850 hPa.



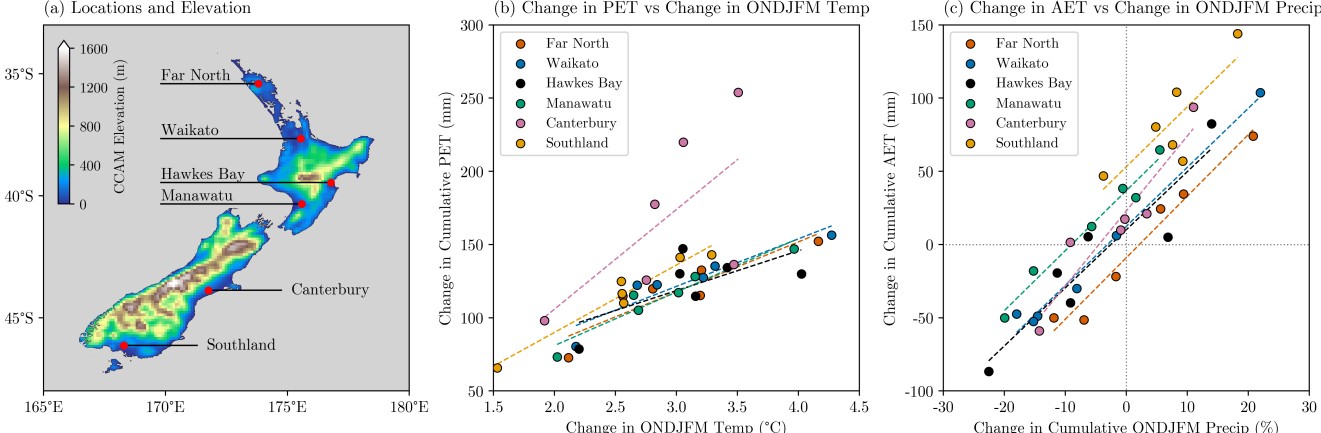

**Figure 4.** (a) Agrarian locations used in this analysis, as well as the elevation of New Zealand gridpoints in the CCAM RCM. (b) Changes in accumulated annual PET across a mean year versus changes in ONDJFM temperature between historical (1985-2014) and future (2070-2099) periods for various agrarian locations around New Zealand. (c) The same as (b) but for changes of annual AET versus ONDJFM rainfall. Each colour represents one location, with each circle representing a downscaled model in our ensemble (n=6), with the dotted line being the line of best fit across all models for each location.

Autumn rainfall delay the recovery of the soil moisture state most significantly in Manawatu and the Hawkes bay, with the 135 wintertime SMD value in the Far North, Hawkes Bay, and Canterbury becoming significantly drier in the future.

The severity of droughts in historical and future periods can be simply quantified using PED, whose climate change signal can be decomposed into contributions of both PET and AET. As seen in Figures 2c and 5, there is a differing response of AET between the two storylines. We detail these opposing effects on the change in mean drought severity in Figure 6. In ACCESS-CM2 (panel (a) of Figure 6), rainfall-driven increases in AET (increases in AET provide a negative contribution to 140 PED) significantly offset the effects of the temperature-driven increases in PET on PED accumulation. This effect reduces the change in drought severity by over 50% in the Far North, Waikato, Hawkes Bay, and Canterbury and mostly offsets any changes in PET accumulation in Southland. The opposite is true for CNRM-CM6-1 shown in panel (b), as soil moisture is no longer replenished by additional rainfall during the dry seasons, the consequential decreases in AET contributes to greater PED accumulation across all regions except Southland. Additionally, the temperature-driven increases in PET propagate through to 145 produce soil moisture drought earlier in the Spring/Summer and thus leads to higher PED accumulation (see Figure 7).

Throughout the results presented, we have detailed the mechanisms contributing to future drought in New Zealand for a mean year in both wetting and drying storylines. In Figure 7, we addressed the drought severity for particularly dry years, examining the five driest years in both 30-year historical and future periods, based on the accumulated PED deficit at the end of each June-July water year. For ACCESS-CM2, the five driest years become significantly drier in the Far North, Waikato, 150 Manawatu, and Canterbury, with small changes in the severity of the future five driest years in Hawkes Bay and Southland.



**Figure 5.** First Row/Third Row: Day-of-year mean values of Historical (blue) and SSP370 (red) PET, alongside Historical (green) and SSP370 (orange) values of AET, across the six agrarian locations in this analysis. Second Row/Fourth Row: Day-of-year mean values of Historical (blue) and SSP370 (red) SMD, across the six agrarian locations in this analysis. The ACCESS-CM2 model is depicted in the top two rows, the CNRM-CM6-1 model is depicted in the bottom two rows.





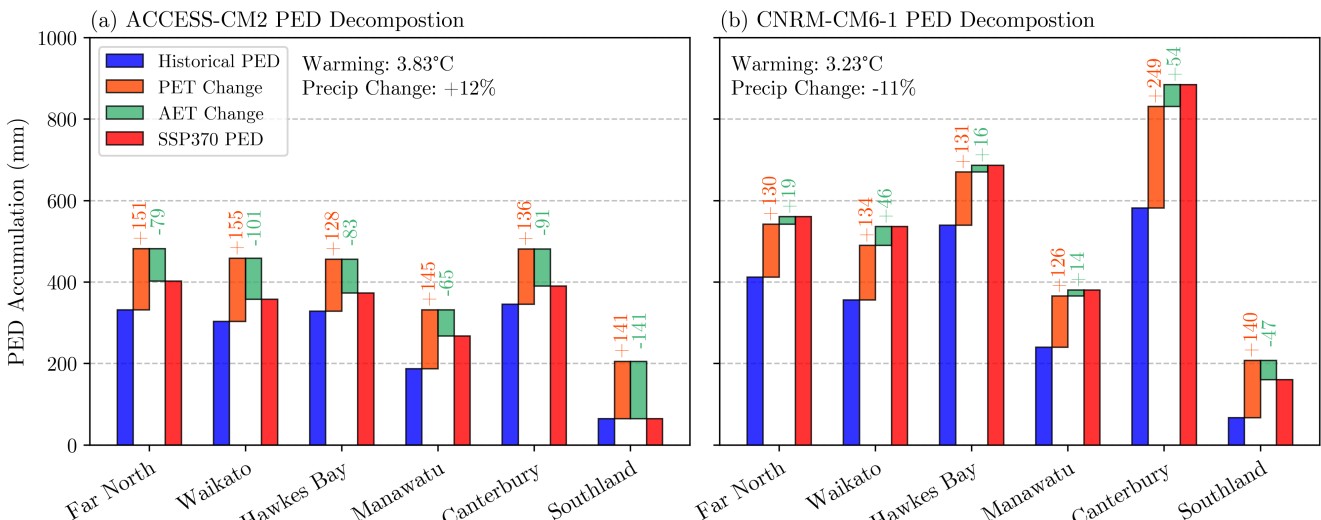

**Figure 6.** Decomposition of changes in PED accumulation at the end of a mean June-July water year from the mean year in the Historical period (blue), to the mean year in the Future period (red) into contributions from changes in PET (orange), and AET(green). Panel (a) depicts these changes for ACCESS-CM2, and panel (b) for CNRM-CM6-1. Numbers above the PET and AET bars indicate the magnitude of their contribution to the change in PED, this means increases in AET provide a negative contribution to PED, while decreases in AET provide a positive contribution.

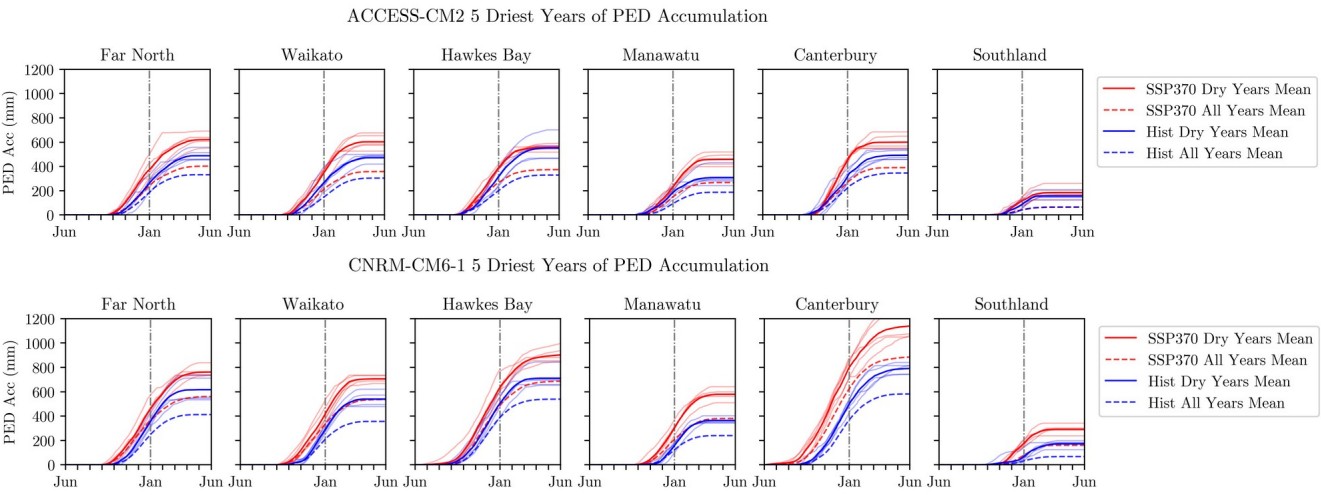

**Figure 7.** The five driest years of PED accumulation (light lines) and the mean of those years (solid lines) in both Historical (blue) and Future (red) periods, for ACCESS-CM2 and CNRM-CM6-1. Dotted lines depict the mean PED accumulation throught the mean year in both historical and future periods.





We see a more modest increase in PED accumulation during a mean year across all regions, apart from Southland where the mean PED accumulation does not change. The increases in PED accumulation in the five driest years in the CNRM-CM6-1 model are larger than those seen in ACCESS-CM2. These increases occurring across all regions, with PED accumulation beginning sooner due to the reduction in Spring rainfall. More significantly, the PED accumulation during a mean year in the future approaches, and in some cases exceeds the mean of the historically five driest years at each location. This presents a potentially disastrous scenario for New Zealand's ecology and agriculture, implying that a typical year in a future climate may become a drought year relative to the historical climate. This is especially true in the Canterbury region, where the mean PED accumulation increases by 50% in the future period. Additionally, under such a scenario, the driest years in the future would be unprecedented relative to any recorded before.

## 4 Discussion

Fundamentally, the selection of any drought metric must confront a trade-off between meaningfully representing the many processes responsible for producing drought-related impacts, but also ensuring confidence in the ability of our modelling tools to simulate drought projections for the right reasons, which includes being able to identify the physical processes which govern any future change (Erian et al., 2021). Some drought metrics are inherently complex, requiring a multitude of variables to calculate, and often including some empirical grounding based on real-world observations from only several locations worldwide. Other, more simple metrics emphasize certain aspects of drought for which we have confidence in being meaningfully represented by climate models (Ukkola et al., 2020). Our choice of metrics in this analysis was motivated by ensuring consistency with past research on agricultural drought in New Zealand. Our framework also allows a transparent pathway to quantify the sometimes-competing influences of projected rainfall and temperature change under future climate scenarios.

There is also some disagreement in the community on the best way to define evaporative droughts in a changing climate. Offline methods using PET tend to overestimate future droughts when compared to direct climate model outputs (Sheffield et al., 2012; Milly and Dunne, 2016; Swann et al., 2016; Yang et al., 2019). These overestimates stem from inappropriate assumptions of the vegetation response to elevated atmospheric $CO_2$ concentrations (Yang et al., 2019), and the double counting of plant feedbacks on surface variables from earth system model outputs (Swann et al., 2016). In the context of our results, the differences between our two storylines are largely due to the rainfall differences between models rather than different levels of temperature-induced drying, which we demonstrate in Figure 4. In fact, ACCESS-CM2 which represents our wetting storyline, is, on average, 0.6°C warmer than CNRM-CM6-1 in the future period. Additionally, in the agricultural locations (pasture land) which are the focus of our study, the land surface is not covered by significant vegetation, meaning the vegetation response is likely less important. These factors lead us to believe that although this overestimation of PET may be present, it should only result in a minimal difference when comparing storylines.

Due to the number of available downscaled climate projections over New Zealand in this ensemble, there are some limitations to this analysis. Here, we have been primarily focused on mean change, briefly touching on the severity of particularly dry years.



Only so much can be done to quantify future drought extremes in a limited number of model years. In future, using a large initial condition ensemble such as the weather@home ensemble (Black et al., 2016; Harrington et al., 2024) could be useful to compliment the existing analysis by more thoroughly examining internal variability in the context of drought projections. Similarly, using artificial intelligence methods to downscale a large ensemble of climate projections (Rampal et al., 2024), which could be used to better quantify model and internal variability..

## 5 Conclusions

In this work we have investigated two divergent future drought scenarios in New Zealand using using dynamically-downscaled CMIP6 climate projections and a storylines approach. Our two storylines encapsulate two extreme futures where droughts are either mitigated or exacerbated by future warm-season changes in rainfall, which is highly uncertain during the summertime in New Zealand.

In the wetting scenario, warming-driven increases in potential evapotranspiration are offset by increases in rainfall, leaving soil moisture state only slightly drier than the historical period. In the drying scenario, the decrease in rainfall causes the soil moisture state to dry out earlier in the Spring and recover more slowly between Autumn and Winter. The wetting scenario shows a modest increase in drought severity in the future for a mean year, where increases in rainfall mitigate 50% of the drying associated with temperature-driven increases in PET. The drying scenario shows a significant increase in drought severity for a mean year, where decreases in rainfall compound the drying associated with the temperature-driven increases in PET. More significantly, in the drying scenario, the drought severity in a mean year in the future becomes comparable to one of the driest years in the historical period.

Our choice of storylines aims to capture the two extremes of the future drought distribution in very high warming scenarios. However, the true spread of the possible changes in drought could be more moderate than depicted in these two scenarios. We give no weight or recommendation to which particular storyline, or combination of the two, is most likely to play out in the future. Even if it is not the most likely outcome, our analysis indicates that this drying scenario remains physically plausible under a high-emissions pathway and should be considered in future planning and adaptation strategies.

*Code and data availability.* The CCAM climate projection ensemble used in this work was produced by NIWA (Gibson et al., 2024b). Access to core variable outputs from this ensemble can be obtained through a free NeSI account: https://www.nesi.org.nz/services/applyforaccess. A subset of the model ensemble output is also freely available at https://climatedata.environment.govt.nz/. Computations of PET were made using the PYET python package (Vremec et al., 2023).

*Author contributions.* Hamish Lewis generated the PET and soil moisture datasets, contributed to the conceptualization of the work, analyzed the data, produced visualizations, and lead the writing of the manuscript. Luke Harrington, conceptualized the work, and assisted in writing





and editing of the manuscript. Peter Gibson ran the CCAM regional climate model, and contributed to writing and editing of the manuscript. Neelesh Rampal produced the code for the water balance model, and contributed to the writing and editing of the manuscript.

*Competing interests.* The authors declare that they have no competing interests.

215   *Acknowledgements.* The authors acknowledge funding from the Royal Society of New Zealand via the Marsden Fund (Grant ID: MFP-UOW2307), and the New Zealand Ministry for Business, Innovation and Employment via their Endeavour Smart Ideas Fund (Grant ID: UOWX2302).



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
