# Peer review of "Storylines of Future Drought in the Face of Uncertain Rainfall Projections: a New Zealand Case Study"

_EGUsphere, 2025_

## Referee Comment (RC1)

**Review of Lewis et al. "Storylines of Future Drought in the Face of Uncertain Rainfall Projections: A New Zealand Case Study"**

**Summary:**

The authors used results from dynamically downscaled RCM simulations over New Zealand to explore two contrasting "storylines" (wetting or drying) and implications for soil moisture droughts. The paper is interesting, well written and suited for ESD. The paper provides important and valuable information to inform changing drought hazard risk in New Zealand. I recommend revisions and have provided some comments that the authors could consider to strengthen their paper. These relate to clarifying the novelty of the study, placing it in context of the wider literature and an expanded discussion of the uncertainties involved.

**Main comments:**

The authors should provide more motivation and introduction on the storyline approach, including a wider literature search of the approach in New Zealand and its applications elsewhere. Has the storyline approach been applied previously in New Zealand for variables like rainfall and temperature and what did they show? The storyline approach or variants of the approach have been applied to look at meteorological, hydrological and soil moisture droughts elsewhere. A wider literature search should enable better scene setting.

- The selection of individual climate model realisation as "storylines" in this study differ from most other uses of the storyline approach in the literature, which tends to group multi-model ensembles based on a selection of physical climate drivers rather than just picking an individual model realisation (e.g. Zappa and Shepherd 2017; Ghosh et al. 2023; Harvey et al. 2023). The authors could consult and refer to the definition of storylines and references therein in the IPCC AR6 Chapter 10 Box 10.2 (Storylines for constructing and communicating regional climate information). One study the author could refer to that do follow similar methodology is that of van der Wiel et al. (2024) but they group multiple dry-trending/wet-trending models.

- There is also another group of studies focused on "event-based storylines" which samples for specific events/sequences that could lead to high impacts (Sillmann et al. 2023). This concept is similar to what the authors did when sampling for the top five driest years. Just a few example seof event based storylines that have been used to explore droughts in Europe (e.g. van der Wiel 2021; Chan et al., 2023).

Details of the water balance model

- Was the water balance model ran on a grid, at the same resolution as the RCMs? Some evaluation or reference to past literature on the water balance model's ability to simulate historical observed soil moisture droughts would also be useful.

- Was the RCM precipitation, temperature and PET bias corrected before being used to drive the water balance model?

- There should also be some discussion of the simulated soil moisture driven by the different model simulations over the historical period – do they exhibit similar long-term average soil moisture behaviour to observations, where does observed soil moisture (or

modelled soil moisture driven by observations) lie within the range of the GCM's historical period?

Results

- It isn't clear whether the two downscaled GCMs chosen as storylines are at diverging ends of the CMIP range or not. The authors do mention that these GCMs were chosen based on historical performance and spread of ECS but if the aim is to characterize uncertainty in drought changes, it would be useful to know where they lie in terms of the wider CMIP range of change in temperature and precipitation?
- Similar to figure 4, consider adding a figure in the main text showing mean change in SMD/drought metric across all GCMs for the different regions, this would help place the two storylines in context.
- The results show that even in the "wetting" storyline, soil moisture droughts in the summer months would still worsen with future warming for some regions due to temperature induced increases in PET, although severity is offset by increase in rainfall. Could the authors reflect on the robustness of this result, would it remain the same given a different GCM/RCM combination which also show a wetting signal? Is it the case that NZ drought will worsen regardless of storyline? The authors could consider adding a figure on transient changes in annual PED accumulation just to visualise changes in soil drought over time from the two storylines.
- The results section focuses on changes in drought within the two climate models but could be strengthened by relating metrics to actual historical observations. The authors made statements such as "*driest years in the future would be unprecedented relative to any recorded before*", implying that direct comparisons with historical observations were made. Rather, the results show droughts in the historical period of the climate model simulations. Can the authors provide some quantification of how much worse the droughts in the future projections are compared to magnitudes of historical observed (or simulated) droughts?

Discussion

- The authors correctly identify that internal climate variability needs to be more thoroughly examined, and this is not possible with only six models and two storylines. There should be more discussion of the different sources of uncertainties, especially that in the near-term, it is likely that internal variability is the dominant uncertainty and single realisations of climate model simulations under-estimates possible changes in drought risk. Additionally, internal climate variability means it is possible in the current climate for unprecedented droughts to occur given natural variability and SMILEs provide an opportunity to explore those droughts (e.g. for Europe: Suarez-Gutierrez et al. 2023 and Australia: Falster et al. 2024).
- Some discussion of the results' sensitivity to the water balance model chosen would be helpful – impact model related uncertainty can be large and is also a main factor in the cascade of uncertainty. Have there been previous attempts to apply national scale hydrological or land surface models?

**Minor comments:**

The authors used the terms "storyline" and "scenario" somewhat interchangeably throughout the manuscript – a more consistent wording would be clearer.

L22: What is "model internal variability"? I think there are two concepts that are mixed up – 1) internal climate variability, which is indeed irreducible but can be better characterised/explored using large ensembles, and 2) the ability of GCM/RCMs to estimate the full range of internal variability, which could relate to model biases. It would be clearer for readers if the authors can identify which of these they are referring to.

L29: For international readers not familiar with projections over New Zealand, it might be worth briefly commenting about general New Zealand projections on rainfall in CMIP5/6. Is the uncertainty in rainfall changes just uncertainty in the magnitude of change or do models also disagree on the sign of change as well. This would help further motivate why a multi-model mean would not be appropriate.

L38: Some general discussion at the start of this paragraph on the different sources of uncertainties that contribute to uncertain rainfall projections would be helpful to set the scene (i.e. uncertainty in atmospheric circulation response to climate change, model biases, internal climate variability, amongst others).

L84: What were the variables required from the RCM to compute PET?

Figure 4 – which of the dots belong to the two GCMs that was chosen as storylines?

Figure 5 – please clarify in text or caption whether the period for the SSP370 lines here was also averages from 2070-2099?

L154 – consider rephrasing – the sentence seems to suggest that you're comparing PED accumulation in the climate model with the five driest years in the historical observations but from my understanding of the results, you're comparing with the five driest years in the historical period of the climate model simulations.

L200 – similar to comment above, should clarify that comparison is made to the historical period in the climate model simulations, not observations.

L201 – consider deleting "even if it is not the most likely outcome" as it is slightly contradictory after the authors point out that no likelihood should be assigned to the storylines in the sentence before.

L203 – unclear what "combination of the two" would mean as the two storylines were selected to be diverging. Do you mean internal variability here?

Abstract – make clear in the abstract that this paper only considers soil moisture/agricultural droughts and not other forms of drought.

**References:**

Zappa, G. and Shepherd, T.G., 2017. Storylines of atmospheric circulation change for European regional climate impact assessment. Journal of Climate, 30(16), pp.6561-6577.

van der Wiel, K., Lenderink, G. and de Vries, H., 2021. Physical storylines of future European drought events like 2018 based on ensemble climate modelling. *Weather and Climate Extremes*, *33*, p.100350.

van der Wiel, K., Beersma, J., van den Brink, H., Krikken, F., Selten, F., Severijns, C., Sterl, A., van Meijgaard, E., Reerink, T. and van Dorland, R., 2024. KNMI'23 climate scenarios for the Netherlands: storyline scenarios of regional climate change. *Earth's Future*, *12*(2), p.e2023EF003983.

Suarez-Gutierrez, L., Müller, W.A. and Marotzke, J., 2023. Extreme heat and drought typical of an end-of-century climate could occur over Europe soon and repeatedly. *Communications Earth & Environment*, *4*(1), p.415.

Sillmann, J., Shepherd, T.G., Van Den Hurk, B., Hazeleger, W., Martius, O., Slingo, J. and Zscheischler, J., 2021. Event-based storylines to address climate risk. *Earth's Future*, *9*(2), p.e2020EF001783.

Harvey, B., Hawkins, E. and Sutton, R., 2023. Storylines for future changes of the North Atlantic jet and associated impacts on the UK. International Journal of Climatology, 43(10), pp.4424-4441.

Ghosh, R. and Shepherd, T.G., 2023. Storylines of Maritime Continent dry period precipitation changes under global warming. Environmental Research Letters, 18(3), p.034017.

Falster, G.M., Wright, N.M., Abram, N.J., Ukkola, A.M. and Henley, B.J., 2024. Potential for historically unprecedented Australian droughts from natural variability and climate change. *Hydrology and Earth System Sciences*, *28*(6), pp.1383-1401.

Chan, W.C., Arnell, N.W., Darch, G., Facer-Childs, K., Shepherd, T.G., Tanguy, M. and van der Wiel, K., 2023. Current and future risk of unprecedented hydrological droughts in Great Britain. *Journal of Hydrology*, *625*, p.130074.

---

## Referee Comment (RC2)

**Review of "Storylines of Future Drought in the Face of Uncertain Rainfall Projections: A New Zealand Case Study" by Lewis et al. 2025.**

Summary:

In this work, Lewis et al. use six dynamically downscaled regional climate models (RCMs) from the CMIP6 ensemble and use the soil water budget to explore drought in New Zealand. They particularly focus on two models, ACCESS-CM2 and CNRM-CM6-1, which produce contrasting signs of rainfall changes, and explore the two diverging "dying-wetting" storylines. For the "wetting" storyline, they show how the increase in rainfall in ACCESS-CM2 model mitigates the drying associated with the temperature-driven increases in potential evapotranspiration (PET). For the "drying" scenario, they show how in fact the decrease in rainfall further exacerbates the drying due to temperature-driven increases in PET. The paper is generally well-written, well-organized and has a good flow. I have some minor suggestions and comments below to help further improve some of the aspects of the manuscript.

General questions to the authors:

1.  I was wondering why and how the emission scenario used in this study was chosen? In section 1, line 49, and in section 2.1, line 74-75, it is mentioned that a "relatively" high emission scenario was chosen. I was wondering why a more extreme scenario (SSP5-8.5) or a more middle-ground scenario (SSP2-4.5) was not chosen for this study? Perhaps adding one or two sentences in section 2.1 to address this would be useful, as the readers might also find it insightful to learn more about it.
2.  Lines 22-23 are a little confusing. In the context of this manuscript, what dose "model internal variability" mean? By looking at the references used there (Deser et al., 2012; Lehner et al., 2020), I immediately think of "internal climate variability". And to address it, large ensembles have been used by the very references mentioned in that line. So, I am not sure if I fully understand the point made here.
3.  What approach/variables were used to calculate PET? Some details could be provided in section 2.2.
4.  Two models, ACCESS-CM2 and CNRN-CM6-1, were singled out to represent the two diverging storylines. Have you tested an approach in which you use an ensemble of models, similar to Wiel et al. (2024)?
5.  In line 126, the six case-study sub-regions are suddenly mentioned in the text, without any introduction beforehand. Maybe it would be better to address this, even briefly, in the Data and Method section? In fact, until Figure 4a, the readers don't actually get to know what sub-regions were studied.

Minor comments:

Line 107: MSLP is mentioned in the text for the first time→ mean sea level pressure (MSLP)

Line 189: omit one "using"

References:

Deser, C., Phillips, A., Bourdette, V., and Teng, H. (2012): Uncertainty in climate change projections: the role of internal variability, *Climate dynamics*, 38, 527–546. https://doi.org/10.1007/s00382-010-0977-x

Lehner, F., Deser, C., Maher, N., Marotzke, J., Fischer, E. M., Brunner, L., Knutti, R., and Hawkins, E.: Partitioning climate projection uncertainty with multiple large ensembles and CMIP5/6, Earth Syst. Dynam., 11, 491–508, https://doi.org/10.5194/esd-11-491-2020

van der Wiel, K., Beersma, J., van den Brink, H., Krikken, F., Selten, F., Severijns, C., et al. (2024). KNMI'23 climate scenarios for the Netherlands: Storyline scenarios of regional climate change. Earth's Future, 12, e2023EF003983. https://doi.org/10.1029/2023EF003983

---

## Author Response (AR2)

**We thank Dr. Wilson Chan for their suggestions which strengthened the manuscript. Review comments appear in Italics while our responses are in bold.**

*Summary:*

*The authors used results from dynamically downscaled RCM simulations over New Zealand to explore two contrasting "storylines" (wetting or drying) and implications for soil moisture droughts. The paper is interesting, well written and suited for ESD. The paper provides important and valuable information to inform changing drought hazard risk in New Zealand. I recommend revisions and have provided some comments that the authors could consider to strengthen their paper. These relate to clarifying the novelty of the study, placing it in context of the wider literature and an expanded discussion of the uncertainties involved.*

*Main comments:*

*The authors should provide more motivation and introduction on the storyline approach, including a wider literature search of the approach in New Zealand and its applications elsewhere. Has the storyline approach been applied previously in New Zealand for variables like rainfall and temperature and what did they show? The storyline approach or variants of the approach have been applied to look at meteorological, hydrological and soil moisture droughts elsewhere. A wider literature search should enable better scene setting.*

**The storylines approach has been used for New Zealand previously for precipitation (Gibson et al. 2024b). We have referenced this work throughout the introduction but now make a point to mention their storylines approach, see line 44. We have further engaged in the literature based on this comment and the two comments following to better contextualize our approach.**

*• The selection of individual climate model realisation as "storylines" in this study differ from most other uses of the storyline approach in the literature, which tends to group multi-model ensembles based on a selection of physical climate drivers rather than just picking an individual model realisation (e.g. Zappa and Shepherd 2017; Ghosh et al. 2023; Harvey et al. 2023). The authors could consult and refer to the definition of storylines and references therein in the IPCC AR6 Chapter 10 Box 10.2 (Storylines for constructing and communicating regional climate information). One study the author could refer to that do follow similar methodology is that of van der Wiel et al. (2024) but they group multiple dry-trending/wet-trending models.*

**We are aware that our use of storylines is slightly different from the more typical definition. We have used the suggested literature to better contextualize our approach in the introduction. See lines 38-55.**

*• There is also another group of studies focused on "event-based storylines" which samples for specific events/sequences that could lead to high impacts (Sillmann et al. 2023). This concept is similar to what the authors did when sampling for the top five driest years. Just a few examples of event based storylines that have been used to explore droughts in Europe (e.g. van der Wiel 2021; Chan et al., 2023).*

**We have mentioned these references for the exacerbation of extreme European droughts to better contextualize our approach. We introduce these on line 167.**

*Details of the water balance model*

*• Was the water balance model ran on a grid, at the same resolution as the RCMs? Some evaluation or reference to past literature on the water balance model's ability to simulate historical observed soil moisture droughts would also be useful.*

**The water balance model was run on the same grid. We now mention this at the beginning of section 2.2. This model does a good job at capturing observed soil moisture droughts, hence its use for New Zealand's real time drought monitoring system (Mol, 2017). Soil moisture outputs from CCAM downscaled ERA5 have an average correlation coefficient of 0.83 across our regions with SMD observations from VCSN , New Zealand's most comprehensive station product.**

*• Was the RCM precipitation, temperature and PET bias corrected before being used to drive the water balance model?*

**These fields were not bias corrected. Nationwide station observations to bias correct to are only available for precipitation and temperature, but are unavailable for the remainder of variables which are used to calculate PET. Thus, we opt to use the raw RCM outputs to maintain consistency across inputs into the water balance model.**

*• There should also be some discussion of the simulated soil moisture driven by the different model simulations over the historical period – do they exhibit similar long-term average soil moisture behaviour to observations, where does observed soil moisture (or modelled soil moisture driven by observations) lie within the range of the GCM's historical period?*

**Soil moisture climatogies calculated using the water balance model driven by CCAM downscaled ERA5 have good agreement with the historical period within the six**

downscaled GCMs in our ensemble. We have added this figure to the supplementary material. However, we note that agreement between the six downscaled GCMs and downscaled observations across the historical period does not imply that future projections are necessarily valid. We communicate this information to the reader in section 2.2.

*Results*

*• It isn't clear whether the two downscaled GCMs chosen as storylines are at diverging ends of the CMIP range or not. The authors do mention that these GCMs were chosen based on historical performance and spread of ECS but if the aim is to characterize uncertainty in drought changes, it would be useful to know where they lie in terms of the wider CMIP range of change in temperature and precipitation?*

The ECS of the models used in this study are outlined in Gibson et al. (2024a), with the two models which represent our storylines on the higher sensitivity side. We now mention this in the first sentence of the results. In terms of regional precipitation during the warm season, both model uncertainty and internal variability completely obscure the sign of change across the ensemble so it is impossible to place them (Gibson 2024b). For example the ACCESS-CM2 member (r1i1pif1) shown in Gibson 2024b has a completely different pattern of change than the member downscaled in this work (r4i1p1f1). We also talk about this in another comment below and discuss how we communicate this to the reader.

*• Similar to figure 4, consider adding a figure in the main text showing mean change in SMD/drought metric across all GCMs for the different regions, this would help place the two storylines in context.*

We elect to keep the figures as they are, as we are focused on contrasting our two storylines. Similar figures can be found in the supplementary material for those who would like this information.

*• The results show that even in the "wetting" storyline, soil moisture droughts in the summer months would still worsen with future warming for some regions due to temperature induced increases in PET, although severity is offset by increase in rainfall. Could the authors reflect on the robustness of this result, would it remain the same given a different GCM/RCM combination which also show a wetting signal? Is it the case that NZ drought will worsen regardless of storyline? The authors could consider adding a figure on transient changes in annual PED accumulation just to visualise changes in soil drought over time from the two storylines.*

**The model would have to have a very small ECS and significant wetting signal for droughts to not worsen. For example, the PET response of NorESM2-MM shown in Figure S8 (b) alongside the AET response of ACCESS-CM2 shown in Figure 6 (a). We discuss this possibility with the reader on line 158.**

*• The results section focuses on changes in drought within the two climate models but could be strengthened by relating metrics to actual historical observations. The authors made statements such as "driest years in the future would be unprecedented relative to any recorded before", implying that direct comparisons with historical observations were made. Rather, the results show droughts in the historical period of the climate model simulations. Can the authors provide some quantification of how much worse the droughts in the future projections are compared to magnitudes of historical observed (or simulated) droughts?*

**Similarly to our comments below, we were trying to reference those droughts which occurred in the model's historical period. See the comments below for resolution of this error. To the point of quantification of historically observed droughts, we now show in the supplementary material PED climatologies of our 6 downscaled GCMs and CCAM downscaled ERA5. We see that ERA5 PED is very similar to ACCESS-CM2 across these regions meaning that future droughts across either storyline would undoubtedly be more severe.**

*Discussion*

*• The authors correctly identify that internal climate variability needs to be more thoroughly examined, and this is not possible with only six models and two storylines. There should be more discussion of the different sources of uncertainties, especially that in the near-term, it is likely that internal variability is the dominant uncertainty and single realisations of climate model simulations under-estimates possible changes in drought risk. Additionally, internal climate variability means it is possible in the current climate for unprecedented droughts to occur given natural variability and SMILEs provide an opportunity to explore those droughts (e.g. for Europe: Suarez-Gutierrez et al. 2023 and Australia: Falster et al. 2024).*

**We have expanded this section further discussing the role of internal variability, particularly in the context of extreme droughts. We have also included the suggested references as future possibilities for this work.**

*• Some discussion of the results' sensitivity to the water balance model chosen would be helpful – impact model related uncertainty can be large and is also a main factor in the*

*cascade of uncertainty. Have there been previous attempts to apply national scale hydrological or land surface models?*

**National scale hydrological models projections have been made for downscaling done on previous CMIP generations (e.g. Collins et al. (2018) for CMIP5). Changes in river mean flow rates differ significantly between the models analyzed in the late century period between individual models, similarly to our results. Unfortunately, the report on the atmospheric portion of the downscaling (Mullan et al. 2018) only show ensemble means so we get an idea of what drives this difference between the models. To the best of our knowledge, there has been no implementation of a national land surface model to study drought such as Ukkola et al. (2016). We have now addressed the model based uncertainty and brought attention to the previous hydrological modeling in the discussion.**

*Minor comments:*

*The authors used the terms "storyline" and "scenario" somewhat interchangeably throughout the manuscript – a more consistent wording would be clearer.*

**We have now elected to use storyline preferentially over scenario where the two would be interchangeable.**

*L22: What is "model internal variability"? I think there are two concepts that are mixed up – 1) internal climate variability, which is indeed irreducible but can be better characterised/explored using large ensembles, and 2) the ability of GCM/RCMs to estimate the full range of internal variability, which could relate to model biases. It would be clearer for readers if the authors can identify which of these they are referring to.*

**Yes, we misspoke here, we have changed this sentence to clarify our point here: "Additionally, internal climate variability can also introduce an irreducible uncertainty in future climate projections, further obscuring future rainfall trends (Deser et al., 2012; Lehner et al., 2020)."**

*L29: For international readers not familiar with projections over New Zealand, it might be worth briefly commenting about general New Zealand projections on rainfall in CMIP5/6. Is the uncertainty in rainfall changes just uncertainty in the magnitude of change or do models also disagree on the sign of change as well. This would help further motivate why a multi-model mean would not be appropriate.*

**It is the sign of change, which is uncertain, both across models and due to internal variability. See Gibson et al. (2024b). We now communicate this to the reader in this sentence.**

*L38: Some general discussion at the start of this paragraph on the different sources of uncertainties that contribute to uncertain rainfall projections would be helpful to set the scene (i.e. uncertainty in atmospheric circulation response to climate change, model biases, internal climate variability, amongst others).*

**We outline these factors in the first paragraph of the paper which we feel sets the scene adequately for this paragraph.**

*L84: What were the variables required from the RCM to compute PET?*

**We have now included this information at the beginning of section 2.2.**

*Figure 4 – which of the dots belong to the two GCMs that was chosen as storylines?*

**We had these models as different markers in a previous version of this figure. However, this made the plot too visually busy. Knowing which points belong to each model doesn't provide the reader with additional information, as we are aiming to detail the sensitivity of PET and AET across the ensemble.**

*Figure 5 – please clarify in text or caption whether the period for the SSP370 lines here was also averages from 2070-2099?*

**We have now clarified that these are the averages across 2070-2099 within the caption.**

*L154 – consider rephrasing – the sentence seems to suggest that you're comparing PED accumulation in the climate model with the five driest years in the historical observations but from my understanding of the results, you're comparing with the five driest years in the historical period of the climate model simulations.*

**We have clarified here that we are talking about the models historical period.**

*L200 – similar to comment above, should clarify that comparison is made to the historical period in the climate model simulations, not observations.*

**Again, we have clarified here that we are talking about the models historical period.**

*L201 – consider deleting "even if it is not the most likely outcome" as it is slightly contradictory after the authors point out that no likelihood should be assigned to the storylines in the sentence before.*

*We have adopted this suggestion.*

*L203 – unclear what "combination of the two" would mean as the two storylines were selected to be diverging. Do you mean internal variability here?*

**Here we intended "combination of the two" to mean some intermediate state. We have rephrased this to: "We give no weight or recommendation to which particular storyline, or intermediate state between the two, is most likely to play out in the future."**

*Abstract – make clear in the abstract that this paper only considers soil moisture/agricultural droughts and not other forms of drought.*

**We have now clarified this in the manuscript.**

**References**

**Gibson, P. B., Stuart, S., Sood, A., Stone, D., Rampal, N., Lewis, H., Broadbent, A., Thatcher, M., and Morgenstern, O.: Dynamical down-scaling CMIP6 models over New Zealand: added value of climatology and extremes, Climate Dynamics, pp. 1–27, 2024a.**

**Gibson, P. B., Rampal, N., Dean, S. M., and Morgenstern, O.: Storylines for future projections of precipitation over New Zealand in CMIP6 models, Journal of Geophysical Research: Atmospheres, 129, e2023JD039 664, 2024b.**

**Mol, A., Tait, A., and Macara, G.: An automated drought monitoring system for New Zealand, Weather and Climate, 37, 23–36, https://www.jstor.org/stable/26735444, 2017.**

**D Collins, K Montgomery, C Zammit: Hydrological projections for New Zealand rivers under climate change: Ministry for the Environment, NZ, 2018**

**B Mullan, A Sood, S Stuart: Climate change projections for New Zealand: Atmospheric projections based on simulations undertaken from the IPCC 5th Assessment: Ministry for the Environment, NZ, 2018**

**Ukkola, A.M., De Kauwe, M.G., Pitman, A.J., Best, M.J., Abramowitz, G., Haverd, V., Decker, M. and Haughton, N.: Land surface models systematically overestimate the intensity, duration and magnitude of seasonal-scale evaporative droughts. *Environmental Research Letters, 2016***

**We thank this anonymous reviewer for their suggestions which strengthened the manuscript. Review comments appear in Italics while our responses are in bold.**

*Summary:*

*In this work, Lewis et al. use six dynamically downscaled regional climate models (RCMs) from the CMIP6 ensemble and use the soil water budget to explore drought in New Zealand. They particularly focus on two models, ACCESS-CM2 and CNRM-CM6-1, which produce contrasting signs of rainfall changes, and explore the two diverging "dying-wetting" storylines. For the "wetting" storyline, they show how the increase in rainfall in ACCESS-CM2 model mitigates the drying associated with the temperature-driven increases in potential evapotranspiration (PET). For the "drying" scenario, they show how in fact the decrease in rainfall further exacerbates the drying due to temperature-driven increases in PET. The paper is generally well-written, well-organized and has a good flow. I have some minor suggestions and comments below to help further improve some of the aspects of the manuscript.*

*General questions to the authors:*

*1. I was wondering why and how the emission scenario used in this study was chosen? In section 1, line 49, and in section 2.1, line 74-75, it is mentioned that a "relatively" high emission scenario was chosen. I was wondering why a more extreme scenario (SSP5-8.5) or a more middle-ground scenario (SSP2-4.5) was not chosen for this study? Perhaps adding one or two sentences in section 2.1 to address this would be useful, as the readers might also find it insightful to learn more about it.*

**We focus on SSP3-7.0 as opposed to SSP2-4.5 as it represents a higher-emissions scenario where the climate change response can more readily be separated from internal variability. We avoid the use of SSP5-8.5 due to the concerns expressed around its realism (e.g. Hausfather and Peters, 2020). We communicate this to the readers at the end of section 2.1.**

*2. Lines 22-23 are a little confusing. In the context of this manuscript, what dose "model internal variability" mean? By looking at the references used there (Deser et al., 2012; Lehner et al., 2020), I immediately think of "internal climate variability". And to address it, large ensembles have been used by the very references mentioned in that line. So, I am not sure if I fully understand the point made here.*

**Yes, we misspoke here, we have changed this sentence to clarify our point here: "Additionally, internal climate variability can also introduce an irreducible uncertainty in future climate projections, further obscuring future rainfall trends (Deser et al., 2012; Lehner et al., 2020)."**

*3. What approach/variables were used to calculate PET? Some details could be provided in section 2.2.*

**We now outline the variables and methodology used to calculate PET at the beginning of section 2.2.**

*4. Two models, ACCESS-CM2 and CNRN-CM6-1, were singled out to represent the two diverging storylines. Have you tested an approach in which you use an ensemble of models, similar to Wiel et al. (2024)?*

**Wiel et al. (2024) group together models with similar precipitation responses to produce wet/dry variants of each future SSP scenario. Unfortunately, due to limitations in computational resources, only these 6 models were able to be dynamically downscaled at the present time. In our case, the only models which would be appropriate to group to fit our storylines based on precipitation projections would be CNRM-CM6-1 and GFDL-ESM4 as they have the same sign of precipitation change nationwide. We will strongly consider this framework in the future where we are able to produce multiple groups of models with a similar sign of precipitation change.**

*5. In line 126, the six case-study sub-regions are suddenly mentioned in the text, without any introduction beforehand. Maybe it would be better to address this, even briefly, in the Data and Method section? In fact, until Figure 4a, the readers don't actually get to know what sub-regions were studied.*

**We now introduce these regions by name at the end of section 2.2 as you suggest.**

*Minor comments:*

*Line 107: MSLP is mentioned in the text for the first time→ mean sea level pressure (MSLP)*

**This has been amended in the manuscript.**

*Line 189: omit one "using"*

**This has been amended in the manuscript.**

**References**

**Hausfather, Z., & Peters, G. P. (2020). Emissions–the 'business as usual' story is misleading. In: Nature Publishing Group.**